# Red-light-mediated copper-catalyzed photoredox catalysis promotes regioselectivity switch in the difunctionalization of alkenes

Tong Zhang[1], Jabor Rabeah [2,3] & Shoubhik Das [1,4] ✉

Controlling regioselectivity during difunctionalization of alkenes remains a significant challenge, particularly when the installation of both functional groups involves radical processes. In this aspect, methodologies to install trifluoromethane ($-CF_3$) via difunctionalization have been explored, due to the importance of this moiety in the pharmaceutical sectors; however, these existing reports are limited, most of which affording only the corresponding β-trifluoromethylated products. The main reason for this limitation arises from the fact that $-CF_3$ group served as an initiator in those reactions and predominantly preferred to be installed at the terminal (β) position of an alkene. On the contrary, functionalization of the $-CF_3$ group at the internal (α) position of alkenes would provide valuable products, but a meticulous approach is necessary to win this regioselectivity switch. Intrigued by this challenge, we here develop an efficient and regioselective strategy where the $-CF_3$ group is installed at the α-position of an alkene. Molecular complexity is achieved via the simultaneous insertion of a sulfonyl fragment ($-SO_2R$) at the β-position. A precisely regulated sequence of radical generation using red light-mediated photocatalysis facilitates this regioselective switch from the terminal (β) position to the internal (α) position. Furthermore, this approach demonstrates broad substrate scope and industrial potential for the synthesis of pharmaceuticals under mild reaction conditions.

Recently, photoredox catalysis has gained tremendous attention in achieving unique synthetic targets under mild reaction conditions[1]. In most of these cases, short-wavelength light regions ($λ_{max} < 460$ nm) were utilized to achieve these reactions successfully. However, short-wavelength light regions have severe limitations of potential health risks such as photooxidative damage to the retina. Furthermore, they can lead to generating undesired side products and thereby, lower the atom economy of that reaction[2-4]. Additionally, lower penetration power of short-wavelength light regions causes concern for the scale-up of that particular reaction[5]. All these limitations have encouraged scientists to move forward to the longer-wavelength regions such as red light or near-infrared (NIR) regions since these are associated with low health risk factors, generate fewer side products due to their lower energy and have high penetration power in the solution which in turn assist to scale up the reaction[6-10]. In longer-wavelength regions, the photocatalysts will be activated by the low-energy. Consequently, their

[1]Department of Chemistry, University of Antwerp, Antwerp, Belgium. [2]Leibniz-Institut für Katalyse e.V. an der Universität Rostock (LIKAT), Rostock, Germany. [3]State Key Laboratory of Low Carbon Catalysis and Carbon Dioxide Utilization, Lanzhou Institute of Chemical Physics (LICP), Chinese Academy of Sciences, Lanzhou, P. R. China. [4]Department of Chemistry, University of Bayreuth, Bayreuth, Germany. ✉e-mail: shoubhik.das@uni-bayreuth.de

**Fig. 1 | Design of the sulfonyltrifluoromethylation of olefins *via* red light-mediated photocatalysis. a** Selective drug molecules containing trifluoromethyl and sulfonyl groups. **b** Site-selective trifluoromethylation of olefin. **c** The requirements for the control of two distinct radicals. **d** Red light-mediated sulfonyltrifluoromethylation of olefin and optimizations (This work). NFSI, *N*-fluorobenzenesulfonimide; PC, photocatalyst; DCE, 1,2-dichloroethane; rt, room temperature; bpy, 2,2′-bipyridine; 1,10-phen, 1,10-phenanthroline.

corresponding redox windows are narrower, and that, in turn, assists in exercising finer control in chemical processes, permitting only specific reactions to take place under defined conditions. Inspired by this, the groups of MacMillan and Rovis have independently developed inspiring photocatalytic strategies for the activation of aryl azide via red light-mediated photoredox catalysis, which have been utilized for proximity labeling[11,12]. Additionally, the utilization of red light-mediated photocatalysis has been increasingly applied across multiple domains to enhance the control of chemical reactions[13–16]. Thus, it is very clear that red light-mediated photoredox catalysis can uniquely attain many unsolved processes that were impossible by the irradiation of ultraviolet (UV) or blue light and that leads to the growing surge of interest in this field, however, it is imperative to acknowledge that still the applications of red light-mediated strategies in organic synthesis are in the early stage of development.

Difunctionalization of alkenes is a powerful synthetic strategy to attain molecular complexity from readily available starting materials[17–23]. In this approach, simultaneously, two different functional groups are installed across an olefin by the introduction of two new C–C or C–X bonds. Along this direction, tremendous catalytic efforts have been paid to attain molecular complexity to design pharmaceutically relevant compounds[24–50]. However, the simultaneous introduction of the trifluoromethyl (−CF₃) and the sulfonyl

fragment (−SO₂R) via difunctionalization is highly challenging due to the intricate difficulty in circumventing undesired side reactions, therefore, rarely has this challenge been solved in organic synthesis. On the other hand, these two functional groups (−CF₃ and −SO₂R) are highly demanding due to their intrinsic capability to enhance the stability, membrane permeability, and metabolism in bioactive molecules and that is reflected in their wide presence as common pharmaceuticals such as CJ-17493 and eletriptan which are served as an NK-1 receptor antagonist, and as a medication for migraine headaches respectively (Fig. 1a)[51–56]. To the best of our knowledge, only a single report has been published for the simultaneous introduction of these two functional groups across the alkene moiety, however, the position of the −CF₃ group was always in the terminal position (β-position)[51]. Along the same direction, it should be clearly noted that the difunctionalization of alkenes via the introduction of a −CF₃ group has frequently been employed. However, −CF₃ group mainly acted as an initiator via the formation of a radical and was always installed to the terminal (β) position of an alkene (as depicted by the solid frame in Fig. 1b). Followed by this terminal addition, subsequent coupling with other functional groups such as -chloro, -chlorosulfonyl, -amino, -carboxylic acid groups were performed to achieve the difunctionalized products[57–62]. On the contrary, reverse regioselectivity of the −CF₃ group at the internal position (α) in the difunctionalized olefins

(indicated by the dashed frame in Fig. 1b) is very rare, although this will allow the achievement of important pharmaceuticals such as CJ-17493, apinocaltamide and many more. To the best of our knowledge, only the group of Li presented an elegant thermocatalytic strategy by involving copper/*N*-fluorobenzenesulfonimide (NFSI) for the introduction of −CF₃ group at the internal position of an alkene (Fig. 1b)[32]. In this approach, the *N*-centered radical, derived from an electrophilic NFSI, served as an initiator to facilitate the addition to the ·β position of the olefin and the (bpy)Zn(CF₃)₂ complex was employed as a nucleophilic −CF₃ reagent.

In this work, inspired by all this information, we became interested in designing a photoredox system that should install both the −CF₃ and −SO₂R groups simultaneously in alkenes where the −CF₃ group should be positioned at the internal position (α) in the difunctionalized product.

## Results and Discussion
### Reaction design
To effectively control the site selectivity, meticulous design of the photoredox strategy during the coupling of two distinct functional groups is inevitable. In the case of Li's protocol, the approach was distinctly different, as they worked with only one radical (*N*-centered radical) in attaining the difunctionalized products[32]. Specifically, when both the −CF₃ and −SO₂R radicals coexist, the −CF₃ radical demonstrates a higher propensity to attach to the olefin first[39,59]. To overcome this obstacle, we argued to ensure: (1) the formation of the −CF₃ radical should occur to the subsequent formation of −SO₂R radical, which will readily initiate the addition to olefins; (2) we also argued to utilize a copper salt as a catalyst to capture the free −CF₃ radical since copper-based salts are well known for simultaneous cross-coupling reactions by involving −CF₃ radical[27,28]. To fulfill these requirements, we attempted to employ a photocatalyst that should be activated by the red light to attain the sulfonyltrifluoromethylated product (Fig. 1c)[63,64]. The reason behind our rationale to use the red light in our reaction was due to the lower energy of the red light compared to the blue light, photocatalysts activated by the red light are expected to exhibit a narrower redox window, enabling a precisely control of radical generation, thereby should facilitate regioselectivity during the addition of two distinct radicals on alkenes. Owing to the narrower redox window of the red light-activated photocatalyst, it was essential to ensure that the excited state of the photocatalyst (PC*) should undergo reduction solely through the sulfinate salts via reductive quenching pathway[46,64]. The resulting sulfonyl radical should then be added to the alkene, leading to the formation of the desired carbon-centered radical. At last, the desired product will be achieved by the carbon-centered radical and Cu−CF₃ complex via Cu-catalyzed cross-coupling reaction[27,28]. In contrast, we rationalized to avoid the oxidative quenching pathway of the PC* since this would have generated free −CF₃ radical, which would result in the undesired trifluoromethylated side products (−CF₃ group at the terminal (β) position)[39,59]. To accomplish this, the photocatalyst was carefully selected based on the redox potentials of sulfinate salts and −CF₃ reagents and the redox potentials should have fulfilled: $E_{ox}(RSO_2^-) < E(PC^*/PC^{\cdot-})$, $E_{red}(CF_3^+) < E(PC^*/PC^{\cdot+})$ and $E(PC^0/PC^{\cdot-}) < E_{red}(CF_3^+)$ (Fig. 1c).

### Reaction optimization
At the outset of the reaction, 4-vinyl-1,1′-biphenyl (1 equiv.), Os(bptpy)₂(PF₆)₂ (0.8 mol%), NaSO₂Ph (3 equiv.) and TTCF₃⁺OTF⁻ (2 equiv.) were employed as the model substrate, photocatalyst, sulfinate salt and −CF₃ reagent in the presence of copper chloride (CuCl₂, 20 mol%) in dichloromethane (DCM, 0.1 M) to afford the sulfonyltrifluoromethylated product (Fig. 1d, details see Supplementary Table 1–3)[5,63,64]. We carefully chosen these reagents (Os(bptpy)₂(PF₆)₂, sodium benzenesulfinate (NaSO₂Ph) and trifluoromethyl thianthrenium triflate (TTCF₃⁺OTF⁻)) based on their redox potential values to match

with our scientific rationale: $E([Os]^{II*/I})$ = +0.93 V *vs.* Ag/AgCl (3 M KCl), $E([Os]^{II*/III})$ = −0.67 V *vs.* Ag/AgCl (3 M KCl)[5], $E_{ox}(NaSO_2Ph)$ = +0.6 V *vs.* Ag/AgCl (3 M KCl)[59,60], $E_{red}(TTCF_3^+OTF^-)$ = −0.69 V *vs.* Ag/AgCl (3 M KCl))[65]. As expected, the performance of the reaction under these conditions did not generate any trifluoromethylated side products (at the terminal position) and only provided the desired product with 73% of yield. It was also observed that reducing the quantities of NaSO₂Ph and TTCF₃⁺OTF⁻, led to a decrease in the yield of the final product (Fig. 1d, entries 2–3). It was necessary to use the excess quantity of sulfinate salts to ensure the faster oxidation of sulfinate salt to the −SO₂R radical. In addition, due to the lower solubility in DCM, the use of the excess quantity of sulfinate salts was highly necessary as well as the presence of an excess quantity of −CF₃ reagent accelerated the reaction rate[25,63,64]. Furthermore, the addition of ligands such as 2,2′-bipyridine (bpy) and 1,10-phenanthroline (1,10-phen) exerted deleterious effects in the reaction, giving no product under these conditions (Fig. 1d, entries 4–5). We assumed that the presence of ligands occupied the coordination sites for −CF₃ radical or hindered the binding of −CF₃ radical to the Cu-center[27]. To verify the importance of the appropriate −CF₃ reagent, alternative electrophilic −CF₃ sources such as Togni's reagent, Umemoto's reagent, and Cu(CF₃)₃bpy were also applied, albeit substantially lower or negligible yield of the desired product was obtained (Fig. 1d, entries 6–10). The rationale behind this could be ascribed to their unsuitable redox potentials, which did not align with Os(bptpy)₂(PF₆)₂ and consequently, failed to meet the requirements. Furthermore, alternative Cu-salts and solvents were also investigated, but lower or negligible yields of the products were obtained (Fig. 1d, entries 11–13). Finally, control experiments revealed that the presence of the photocatalyst, Cu-salts and red light were essential for this reaction (Fig. 1d, entries 14–16).

In order to exhibit the red light-mediated regioselective gain for this reaction, reaction conditions under the irradiation of blue light were also compared. Similar to the 'red light system', the crucial combination of the photocatalyst, sulfinate salt and −CF₃ reagent was determined, namely [Ru(bpz)₃](PF₆)₂, NaSO₂Ph and 5-(trifluoromethyl)dibenzothiophenium triflate (Fig. 2b). However, after extensive optimizations via the investigation of each crucial component of this reaction, the highest yield of the desired product reached to 42% and this could be due to the fact that free −CF₃ radical was generated faster under these conditions (see SI 1.3.2). In addition, the generation of free −CF₃ radicals could also be attributed to the more powerful blue light. Subsequently, the −CF₃ radical underwent an addition reaction with styrene, resulted in the formation of the undesired β-substituted trifluoromethylated byproduct and the contrast was notably evident in the ¹⁹F NMR spectra (Fig. 2c). The 'blue light system' exhibited numerous peaks of side products while the spectrum of the 'red light system' appeared significantly cleaner and mainly contained the −CF₃ reagent and the desired product. This significant difference highlighted the pronounced regioselectivity gain in the sulfonyltrifluoromethylation of alkenes via the red light-mediated photocatalysis.

### Substrate scope
With these optimized reaction conditions in hand, we started to evaluate the scope of the sulfonyltrifluoromethylation of alkenes. As shown in the Fig. 3, an array of *para*-substituted styrenes containing diverse electron-donating groups (EDGs) like -methyl, -acetoxy, and -*tert*-butyl, as well as electron-withdrawing groups (EWGs) such as -halogens provided the corresponding sulfonyltrifluoromethylated products in moderate to excellent yield (Fig. 3, **1**–**8**). Specifically, 4-bromostyrene and 4-chlorostyrene were tolerant under our optimized conditions to provide the desired products (**6** and **7**), thereby, demonstrated the potential for subsequent functionalization via cross-coupling reactions[32]. Furthermore, the reaction demonstrated compatibility with 2- and 3-substituted styrenes (**10**–**13**), leading to the formation of products in satisfactory yield, regardless of the presence

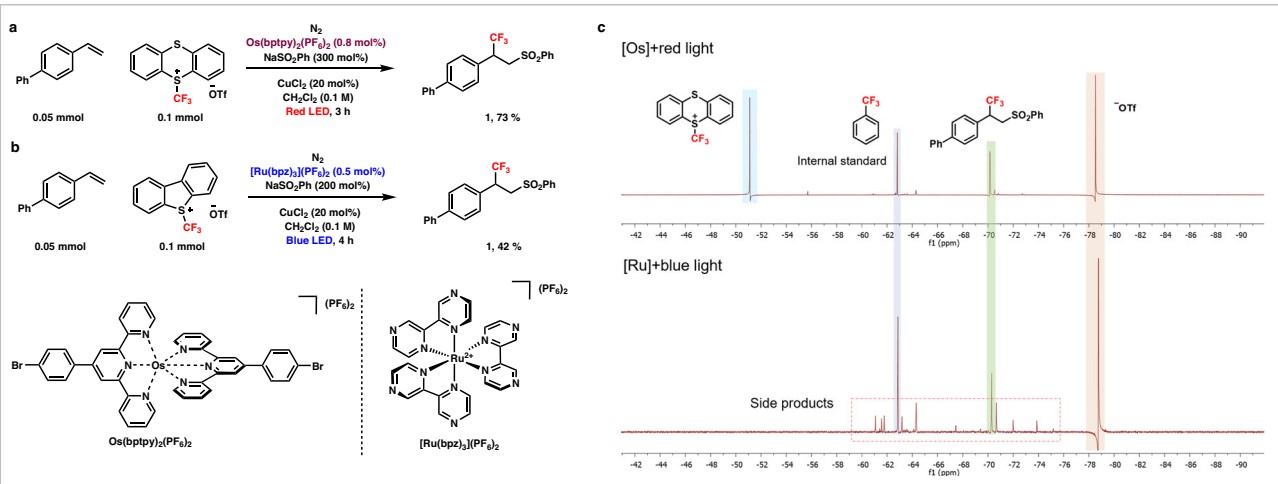

**Fig. 2 | Comparison of the reaction under blue and red light with respective photocatalysts. a** Reaction under optimized conditions with the irradiation of red light. **b** Reaction under optimized conditions with the irradiation of blue light. **c** $^{19}$F NMR spectra of the reaction under blue and red light conditions.

of -EDGs or -EWGs. In comparison, electron-deficient alkenes (**9** and **14**) exhibited decreased efficiency, however, the use of *p*-chlorophenyl sulfinate led to an improvement in the reaction. In general, the difunctionalization of β-substituted styrenes represents increased difficulty due to the hindrance caused by these β-substituents and this hindrance can impede the addition of initiators, such as sulfonyl radicals in this work[32]. However, under our optimized reaction conditions, (*E*)-β-methylstyrene (**15**) and indene (**16**) underwent the difunctionalization reaction smoothly and provided a yield of 46% and 78%, respectively. However, unactivated alkenes have not successfully yielded the desired sulfonyltrifluoromethylated products (see SI 1.4.5).

Encouraged by these results, an extensive exploration of sulfinate salts was conducted within the optimized reaction conditions. To our delight, a diverse array of *p*-substituted phenyl sulfinates, encompassing -methyl, -chloro, -bromo, -nitro, and -cyano groups, demonstrated excellent tolerance, yielding the desired products in yields from good to excellent (**17-21**). Furthermore, aliphatic sulfinates (**22** and **23**) also proved to be compatible which exhibited strong application potentials in pharmaceutical area such as the modification of azidothymidine which is known as an anti-HIV drug[66]. The adaptability of our methodology extended further to sulfinates bearing biphenyl-, cyclopropane-, and thiophene-groups. These substrates smoothly underwent difunctionalization reactions under the irradiation of red light, yielding products in the range of 35-93% (**24-26**). This exhibited wide generality of our system to afford various sulfones-containing chemicals, thereby making significant contributions to the field of pharmaceuticals, agrochemicals, and it should be also noted that the synthesis of sulfones-containing chemicals is of paramount importance in organic chemistry[46-48].

Recently, the focus on late-stage modification has garnered significant interest due to its direct and efficient approach in synthesizing functionalized complex molecules[67-71]. The expedite synthesis of highly-functionalized molecules holds strong promise for its potential utility in various scientific disciplines including drug discovery, materials science, and molecular imaging[71]. To evaluate the application of our method on complex molecules, a series of drug molecules and natural products derivatives such as estrone, (*S*)-(+)-naproxen, dexibuprofen, (1*S*)-(−)-camphanic acid, indomethacin and adapalene were applied (**27-32**). Under our experimental conditions, these diverse drug derivatives, encompassing a variety of functional groups, exhibited excellent tolerance and compatibility. The resulting products were obtained in yields from 66% to 88%, indicating high reaction efficiency. This demonstrated the potential of our methodology in facilitating the synthesis of more complex sulfonyltrifluoromethylated

molecules. We strongly believe that the -trifluoromethyl and -sulfonyl groups in functionalized drug molecules and natural products should not only improve their inherent properties but should also provide the opportunity for further transformation.

## Application potentials

To further examine the application potential, a 4 mmol-scale reaction was carried out which proceeded smoothly in 4 hours and yielded 0.85 grams of the desired product (Fig. 4a). In addition, product **6** synthesized from 1-bromo-4-vinylbenzene could smoothly give **33** with *p*-tolylboronic acid via Suzuki-coupling reaction (Fig. 4b)[72]. Due to the superior light penetration of red light, it became feasible to directly conduct the upscaling of the reaction within a batch reaction system[5]. To further demonstrate the synthetic utility of our strategy, the elimination of the -sulfonyl group was achieved through a straightforward strategy by using a mixture of Cs$_2$CO$_3$ and 7-methyl-1,5,7-triazabicyclo(4.4.0)dec-5-ene (MTBD), resulting in the production of α-trifluoromethyl styrene (**34**) with a yield of 90% (Fig. 4c)[64]. The mixture of base facilitated the deprotonation and desulfonylation of the sulfonyltrifluoromethylated styrenes to form the α-trifluoromethyl styrenes. In general, α-trifluoromethyl styrene derivatives are highly important as versatile synthetic intermediates for the construction of complex fluorinated compounds, which are synthesized through methylenation of trifluoromethylketones (Wittig reaction) or via transition metal-catalyzed cross-coupling reactions[73,74]. However, compared to these approaches, our strategy enabled the direct synthesis of α-trifluoromethyl styrene derivatives from styrene, eliminating the requirement of Wittig reagents as well as -borylated or -halide reagents in the processes to improve the atom economy. Additionally, the obtained α-trifluoromethyl styrene was further transformed into *gem*-difluoroalkenes (**35**) in 86% yield and these fluorinated compounds have strong potential to act as a ketone mimic in pharmaceuticals[75-77]. In fact, substitution of the carbonyl group by the *gem*-difluoroalkene moiety has been shown to enhance the oral bioavailability of therapeutic agents[75]. Furthermore, our strategy generated a key intermediate (**36**) for the synthesis of apinocaltamide (**38**), T-type calcium channel blocker from 4-bromostyrene (Fig. 4d)[78,79]. All these approaches clearly demonstrate the strong potential of our strategy for further applications in designing or modifying pharmaceuticals.

## Mechanistic investigations

Inspired by all these outcomes, we became interested in validating the reaction mechanism of this unique reaction strategy and a series

**Fig. 3 | Scope of the sulfonyltrifluoromethylation of olefins[a].** [a]Yields are reported as isolated yield. [b]dr value was determined by [1]H NMR.

of mechanistic experiments were conducted to validate our mechanistic proposal (Fig. 5). At first, (2,2,6,6-Tetramethylpiperidin-1-yl)oxyl (TEMPO) was added as a radical quenching reagent under the optimized reaction conditions. As expected, a trace quantity of the product was obtained and a carbon-centered radical (**III**) was captured by TEMPO which was detected by the high-resolution mass spectrometry (HRMS) (Fig. 5a), indicating that the radical process was involved. To further support the involvement of radicals during the addition of the sulfonyl radical, a radical probe experiment was conducted where the model styrene (**40**) yielded the ring-opening product **41** (Fig. 5b). Upon the addition of sulfonyl radical to **40**, a cyclopropylmethyl radical moiety was formed, followed by the rapid ring opening rearrangement relieved the ring strain and finally, resulted in the final ring-opening product (**41**). Additionally, Stern−Volmer fluorescence quenching experiments were conducted, revealing that the sodium sulfinate salt exhibited the highest potential as a quencher for the excited state of the Os-photocatalyst, which was also corroborated by the electrochemical measurements for redox potentials (Fig. 5c, see SI 1.4.1)[5]. In Fig. 5c, it is demonstrated

that as the concentration of sulfinate salt was increased, there was a notable reduction in fluorescence intensity. However, minimal alterations were detected in the case of the −CF₃ reagent, styrene, and CuCl₂. This observation was aligned with the anticipated reductive quenching pathway and supported our design that the generation of ·sulfonyl radical was prior to the generation of −CF₃ radical in the reaction, indicating that no free −CF₃ radical was generated and ensuring the high regioselectivity switch in this reaction. Furthermore, the formation of Cu−CF₃ active species was also investigated and to analyze the possible Cu-CF₃ active species, various control experiments were carried out (Fig. 5d). Initially, we attempted to detect the active species in the absence of styrene under model reaction conditions. No new peak corresponding to Cu$^{II}$−CF₃ was observed in 1–4 h, however, we observed the presence of the Cu$^{III}$(CF₃)₄ anion peak (Fig. 6a). Due to the potential instability of the Cu$^{II}$−CF₃ complex, we further attempted the addition of the bpy ligand to detect the potential existence of the Cu$^{II}$−CF₃ in Fig. 6a. However, only peak of TTCF₃$^+$OTF$^-$ was observed in [19]F NMR (Fig. 6b). The presence of ligands either occupied the available coordination

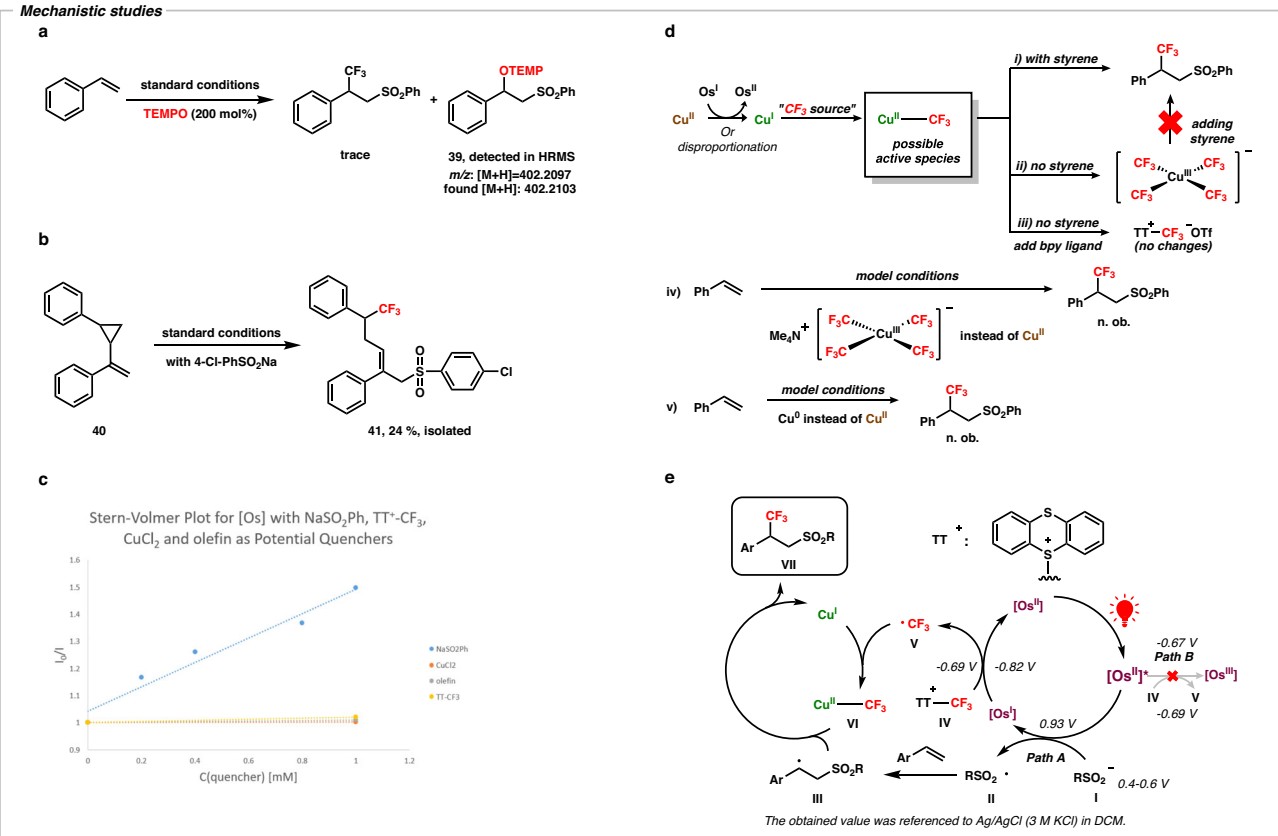

**Fig. 4 | Post-functionalization of the sulfonyltrifluoromethylated product. a** Gram scale reaction. **b** Suzuki-coupling reaction. **c** Elimination of sulfonyl group and followed by defluorination. **d** Key intermediate generation for synthesis of Apinocaltamide. DMF, dimethylformamide; MTBD, 7-methyl-1,5,7-triazabicyclo(4.4.0)dec-5-ene.

**Fig. 5 | Mechanistic studies. a** Quenching experiments with TEMPO. **b** Radical probe experiment via ring-opening reaction. **c** Fluorescence quenching experiments. **d** Analysis of Cu−CF₃ active species. **e** Proposed mechanism of this work. TEMPO, (2,2,6,6-Tetramethylpiperidin-1-yl)oxyl; bpy, 2,2′-bipyridine.

sites of −CF₃ radical or impeded the binding of −CF₃ radical to the Cu-center[27]. To further verify the Cu$^{III}$(CF₃)₄ anionic complex, we synthesized stable Me₄NCu$^{III}$(CF₃)₄ complex by following the reference article[80]. However, no product was obtained by using Me₄NCu$^{III}$(CF₃)₄ complex instead of CuCl₂ under our optimized reaction conditions (Fig. 6c). Similarly, to verify the possibility of Cu$^{I}$−CF₃ complex as active species, the model reaction was carried out by replacing CuCl₂

with fresh copper powder (Cu⁰) and as expected, no product was obtained under this condition (Fig. 6d). By analyzing all these experiments, we could assume that the active species Cu−CF₃ were not in the form of Cu$^{III}$−CF₃ or Cu$^{I}$−CF₃ complexes but possibly were in the form of Cu$^{II}$−CF₃ complex, which was also corroborated by the electron paramagnetic resonance (EPR) analysis of reactions (see Supplementary Fig. 9).

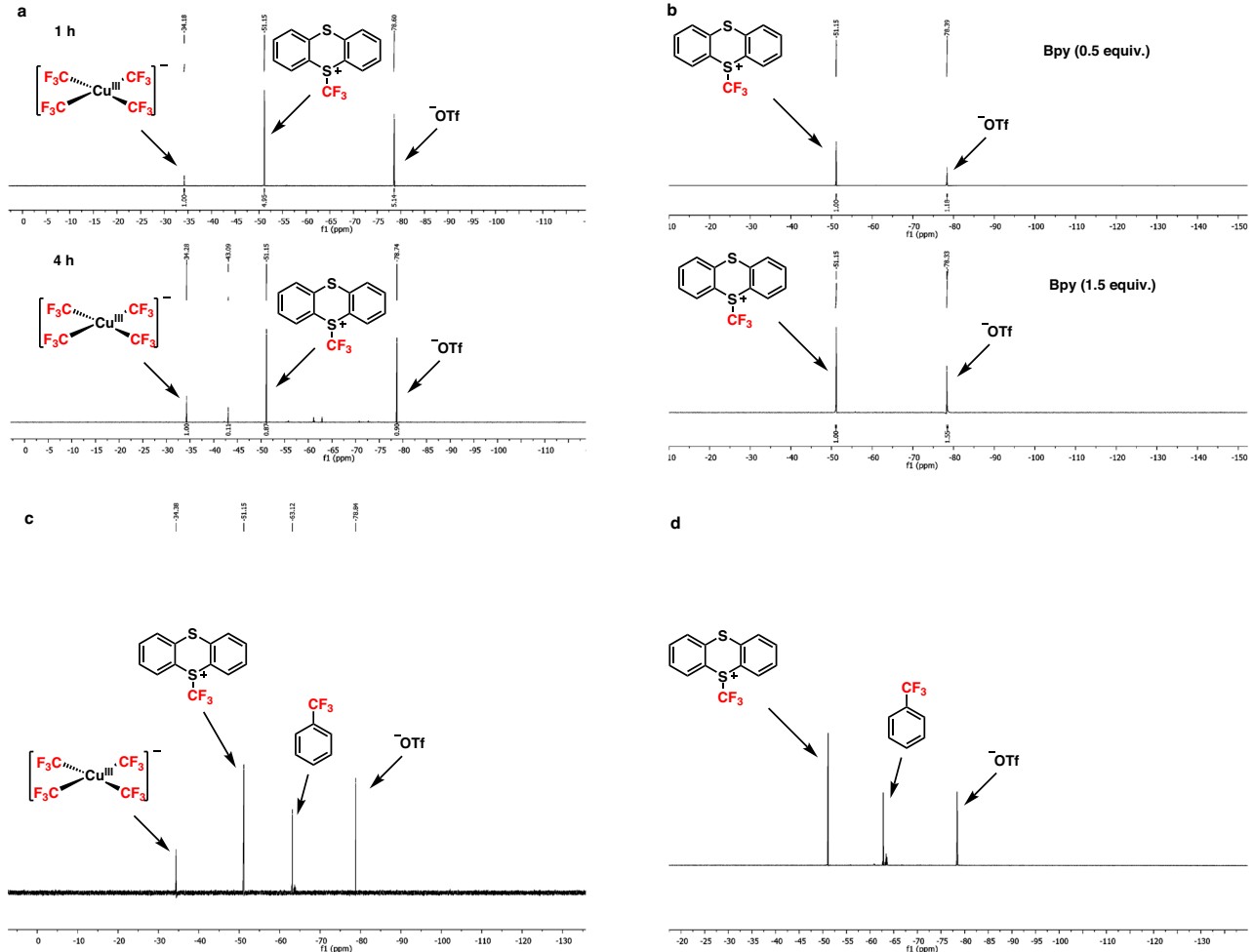

**Fig. 6 | NMR spectra of the analysis for Cu−CF₃ complex. a** Model reaction in the absence of styrene after 1 h and 4 h. **b** Experiment A with the addition of bpy (0.5 or 1.5 equiv.) as ligand. **c** Model reaction by replacing CuCl₂ with Me₄NCu^III(CF₃)₄ complex. **d** Model reaction by replacing CuCl₂ with fresh Cu powder. bpy, 2,2′-bipyridine.

Based on all these mechanistic studies, we proposed a possible mechanism for the overall reaction system (Fig. 5e). The excited state of the photocatalyst [Os^II]* ($E^{II*/I}$ = +0.93 V *vs*. Ag/AgCl (3 M KCl), $E^{II*/III}$ = −0.67 V *vs*. Ag/AgCl (3 M KCl))[5] was activated by the red light and exclusively underwent reduction by the sulfinate salts, **I** ($E_{ox}$ = +0.4 − 0.6 V *vs*. Ag/AgCl (3 M KCl))[63,64] to form the sulfonyl radical **II** (Path A) rather than oxidation by TTCF₃⁺OTF⁻ **IV** ($E_{red}$ = −0.69 V *vs*. Ag/AgCl (3 M KCl))[65] to generate the free −CF₃ radical **V** (Path B), which was consistent with the result of fluorescence quenching experiments. The formed sulfonyl radical **II** was added to the alkene to generate a carbon-centered radical **III,** which was verified by the TEMPO quenching experiment and the radical probe experiment. Later, the Cu^I-species captured the free −CF₃ radical **V**, generated through the reduction of **IV** by [Os^I] ($E^{II/I}$ = −0.82 V *vs*. Ag/AgCl (3 M KCl))[5], resulted in the formation of the Cu^II−CF₃ complex **VI**. At last, the final product **VII** was delivered via the cross-coupling reaction between **III** and **VI**. In addition, this reaction process is a closed catalytic cycle according to the calculation of quantum yield (see SI 1.4.4).

In summary, we have developed a unique protocol where red light-mediated photocatalysis triggered a regioselective switch during the sulfonyltrifluoromethylation of olefins. This strategy has effectively addressed the challenges associated with regioselective addition of radicals onto alkenes. The broad substrate scope and late-stage transformation demonstrated the high efficiency of these reactions and also proved the excellent tolerance of functional groups.

Furthermore, post-functionalization studies highlighted the significant industrial potential of the sulfonyltrifluoromethylated product. Additionally, detailed mechanistic investigations revealed a sequential generation of radicals, followed by Cu-catalyzed cross-coupling reactions. We believe that this strategy will strongly contribute to the regioselective functionalizations and will further inspire the development of additional methods in this field.

## Methods

### General procedure for sulfonyltrifluoromethylation of olefins

A dried reaction vial with a magnetic stirring bar was charged with Os(bptpy)₂(PF₆)₂ (0.0008 mmol, 0.8 mol%), CuCl₂ (0.02 mmol, 20 mol %), TT−CF₃⁺OTF⁻ (0.2 mmol, 2 equiv.) and sodium sulfinate (0.3 mmol, 3 equiv.). After charging all these reagents, the vessel was evacuated by using Schlenk techniques and flushed with N₂ three times. Under nitrogen gas flow, olefin (0.1 mmol, 1 equiv.) (if liquid, otherwise added before flushing cycle) and dry DCM (0.1 M) were added by using a syringe which was flushed with inert gas. The resulting mixture was stirred for 3–4 h under the irradiation of red LED light (EvoluChem™ LED 650PF HCK1012-XX-014 650 nm 20 mW/cm²) in the EvoluChem PhotoRedOx Box. After the completion of the reaction, the reaction mixture was quenched by adding distilled water (2 mL). The organic phase was extracted and concentrated in vacuo. 1,1,1-Trifluorotoluene was added as an internal standard to determine the NMR yield of the functionalized product through ¹⁹F NMR. Purification proceeded via flash column chromatography.

## Data availability

The materials, reaction optimization, mechanism investigation, general procedure of reactions, characterization of substrates and spectra of products, as well as all other supporting data generated in this study are provided in this manuscript and in the Supplementary Information. Any additional data that support the findings of this study are available from the corresponding authors upon request.

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

## Acknowledgements

S.D. thanks the Francqui start up grant from the University of Antwerp, Belgium, for the financial support. T.Z. thanks FWO SB PhD fellowship for their financial assistance to finish this work. We thank Dr. Rakesh Maiti from University of Bayreuth for helpful discussions. We also thank Mr. Glenn Van Haesendonck from UAntwerpen, Belgium for HRMS measurements.

## Author contributions

T.Z. and S.D. designed the project. T.Z. developed the reaction, investigated the substrate scope, examined the applications, and studied the reaction mechanism. J.R. did an EPR analysis. Finally, T.Z. and S.D. wrote the manuscript.

## Funding

## Competing interests
The authors declare no competing interest.
