## [Peer Review File · Nature Communications]

REVIEWER COMMENTS

Reviewer #1 (Remarks to the Author):

In this manuscript, Zhang and Das report the use of a Os based photocatalyst for the red light mediated difunctionalization of alkenes. Remarkably, the authors report a methodology in which CF₃ is added at the a-position of the styrene instead of b-position normally observed in a radical pathway. To achieve this regioselectivity, the authors have carefully designed a methodology in which the photocatalyst thermodynamic parameters force the system to add CF₃ radical as the second step of alkene functionalization. By using a photocatalyst with specific redox potential, the authors inhibit the oxidative quenching pathway which is normally responsible for the CF₃ radical formation. Due to redox potential mismatch, the Os photocatalyst undergoes a reductive quenching pathway where NaSO₂Ph is oxidized to the SO₂Ph radical prior the generation of CF₃ radical, resulting in a reverse regioselectivity where SO₂Ph is first added to the b-position, while CF₃ radical adds to the a-position. In addition, the authors have reported a large scope of reaction supporting the versatility of this methodology. Finally, they performed a mechanistic study that allow them to write a plausible mechanism.

This is a very exciting reports, that show how regioselectivity can be obtain by choosing a photocatalyst with the right potential, and that red light photocatalyst are suited thanks to their narrow range of redox potentials and the use of low energy light. I only have minor comments and suggestions:

1) In the mechanistic study, the author talk about the lack of Cu(II)-CF₃ species observed in NMR. But Cu(II) is a d⁹ species that will be paramagnetic and not observe by NMR. I suggest to rephrase that paragraph so it doesn't look like the author were expecting to observe a signal. I would also suggest to perform EPR spectroscopy on the 4 experiments to rule out or confirm the formation of CuII-CF₃ species. I understand that with the Os present, EPR maybe challenging to interpret, but it needs to be done.

2) In the propose mechanism, Cu(I) is the resting state of Cu, which then is turn into Cu(II)-CF₃, what is the initiation step to convert the CuCl₂ to the active Cu(I). Does the OsII*/I photoredox potential can reduce Cu(II) to Cu(I)? if so, why did the stern-volmer show no quenching in presence of CuCl₂?

3) Also in the mechanistic study, where does the PhCF₃ come from in experiments C and D?

4) The author use $[\text{Ru}(\text{Bpz})_3][\text{PF}_6]_2$ as PC for blue light to highlight the advantage of red light. It would be useful to add, in the text or ESI, the redox potential of this PC compared to the reactant (similar to Figure 1C). Is the lack of regioselectivity due to the higher energy blue light or simply because the thermodynamic parameters of the Ru PC don't match as well. Along that line using, another Red-light PC that have different redox potential could be interesting to look at in the future (for example DMQA+ see: J. Am. Chem. Soc. 2020, 142, 28, 12056–12061 and J. Am. Chem. Soc. 2024, 146, 12, 7922–7930, these articles could be cited in the intro with ref 6-9 for the first one and 12-14 for the second).

5) Minor typo:

- In Figure 1, NaSO_2Ph is shown in equiv., while everything else is in mol. Then in Figure 2, NaSO_2Ph is shown in mol%. I would suggest to be consistent and add mol% in Figure 1.

- Text in the mechanistic study need some review, for example:

o "In Figure 5c, it demonstrated..." should be "In Figure 5c, it is demonstrated..."

o "Furthermore, the form of Cu-CF₃ species ..." should be "Furthermore, the formation of Cu-CF₃ species ..."

o "Initially, we attempted to detect the active species in the absence of styrene under model reaction conditions, while no new peak corresponding to CuI-CF₃ was observed in 1 - 4 h, however, we observed the presence of the CuIII(CF₃)₄ anion peak" may read better as "Initially, we attempted to detect the active species in the absence of styrene under model reaction conditions. No new peak corresponding to CuI-CF₃ was observed in 1 - 4 h, however, we observed the presence of the CuIII(CF₃)₄ anion peak"

Reviewer #2 (Remarks to the Author):

The report by Das and coworkers details a practical platform for regioselective sulfonyltrifluoromethylation of alkenes by leveraging red light-mediated photoredox/copper catalysis, enabling the installation of the CF₃ group at internal positions of alkenes. This is an interesting approach using red light to reverse regioselectivity, complementing the well-known regioselectivity of radical-based trifluoromethyl-functionalization of alkenes (Angew. Chem., Int. Ed. 2015, 54, 6999). This chemistry shows a good substrate scope and synthetic applications. Overall, this is a good addition to the red light-mediated photoredox catalysis, although the scope of this

chemistry could be further expanded. This reviewer thinks this paper would merit its publication in Nature Communications only if the authors can address the following issues:

1, Unactivated alkenes are amenable substrates in Li's aminotrifluoromethylation reactions (J. Am. Chem. Soc. 2019, 141, 29, 11440). However, this paper only shows the reactivity of styrenes. Have the authors tried unactivated alkenes?

2, Although introducing the sulfonyl group is important, have the authors tried other radical precursors? Can we expect similar reactivity if the redox potential of a reagent falls within the window shown in Figure 1? It's important to let the audiences know this approach's synthetic potential.

3, In Figure 2, it is not meticulous enough to conduct the parallel reactions with only Umemoto's reagent under blue light, as CF₃-source also plays an important role in this reaction (see in Table S2 and part 1.3.2 in SI).

4, The synthetic utility, which is mainly based on α -trifluoromethyl styrene via elimination of the sulfonyl group, is not very convincing, as α -trifluoromethyl styrene can be easily synthesized via Wittig reaction of trifluoromethylketones. More interesting applications are suggested to be included.

5, The Authors proposed an out-of-cage CF₃ radical in Figure 5e. This is consistent with its unique regioselectivity, as CF₃ radicals are prone to adding to styrenes. More experiments and explanations on the selectivity should be provided.

6, It would be more appropriate to reflect the involvement of copper catalysis in the title.

Reviewer #3 (Remarks to the Author):

Das et al described an efficient method for the synthesis of sulfonyltrifluoromethylated products through Red Light-Mediated Photoredox Catalysis using Os(btpy)₂(PF₆)₂ as a photoredox catalyst. A series of sulfonyltrifluoromethylated products were synthesized in good yields. The manuscript is well-written and adequate experiments were carried out to support the proposed mechanism. The manuscript can be acceptable for publication after addressing the following queries.

1. Is it a radical chain process? Therefore the quantum efficiency of the reaction must be determined as recently described by Yoon and Stephenson (*Science*, 2015, 349, 1285).
3. TEMPO was added to intercept the radical reaction. If so, the yield and structure of the adduct should be given.
4. The mechanistic proposal provided by the authors, requires better experimental support, such as Stern-Volmer studies.

REVIEWER COMMENTS

Reviewer #1

In this manuscript, Zhang and Das report the use of a Os based photocatalyst for the red light mediated difunctionalization of alkenes. Remarkably, the authors report a methodology in which CF₃ is added at the a-position of the styrene instead of b-position normally observed in a radical pathway. To achieve this regioselectivity, the authors have carefully designed a methodology in which the photocatalyst thermodynamic parameters force the system to add CF₃ radical as the second step of alkene functionalization. By using a photocatalyst with specific redox potential, the authors inhibit the oxidative quenching pathway which is normally responsible for the CF₃ radical formation. Due to redox potential mismatch, the Os photocatalyst undergoes a reductive quenching pathway where NaSO₂Ph is oxidized to the SO₂Ph radical prior the generation of CF₃ radical, resulting in a reverse regioselectivity where SO₂Ph is first added to the b-position, while CF₃ radical adds to the a-position. In addition, the authors have reported a large scope of reaction supporting the versatility of this methodology. Finally, they performed a mechanistic study that allow them to write a plausible mechanism.

This is a very exciting reports, that show how regioselectivity can be obtain by choosing a photocatalyst with the right potential, and that red light photocatalyst are suited thanks to their narrow range of redox potentials and the use of low energy light. I only have minor comments and suggestions:

Response: We appreciate reviewer's positive evaluation about our work as well as the constructive feedback for the further improvement of the manuscript. The responses to reviewer's comments are provided below.

1) In the mechanistic study, the author talk about the lack of Cu(II)-CF₃ species observed in NMR. But Cu(II) is a d9 species that will be paramagnetic and not observe by NMR. I suggest to rephrase that paragraph so it doesn't look like the author were expecting to observe a signal. I would also suggest performing EPR spectroscopy on the 4 experiments to rule out or confirm the formation of Cu(I)-CF₃ species. I understand that with the Os present, EPR maybe challenging to interpret, but it needs to be done.

Response: We appreciate the comment and suggestion from the reviewer regarding the detection of Cu(II)-CF₃ species. The EPR analysis of reactions has been done and shown below. We have also added that information in the SI.

Reaction conditions: CuCl₂ (2.7 mg), TTCF₃⁺OTF⁻ (9.5 mg), NaSO₂Ph (5.9 mg) and Os(bptpy)₂(PF₆)₂ (0.6 mg) in 1 mL DCM in dark and under the irradiation at room temperature; a) without olefin; b) in the presence of 5.1 mg model olefin.

The EPR spectrum of the reaction mixture recorded in dark is characterized by three g values ($g_x = 2.244$; $g_y = 2.187$; $g_z = 2.047$). The g values ($g_x > g_y > g_z$) are characteristic of pentacoordination Cu^{II} ion with a geometry intermediate between the square pyramid and trigonal bipyramid. Illuminating the

mixture resulted in a decreasing of the Cu^{II} EPR signal (**Scheme S9**) due to the formation of EPR silent species, probably to Cu^I as suggested in **Figure 5d**. However, this effect is less pronounced in the presence of olefin, more probably due to reoxidation of Cu^I to Cu^{II} during the photocatalytic reaction. It seems to be that the reduction step of Cu^{II} to Cu^I is higher than reoxidation of Cu^I to Cu^{II} as the signal of Cu^{II} decreased with time.

Based on the EPR and NMR analysis, we proposed that Cu^{III}(CF₃)₄⁻ complex was formed in the absence of the olefin, which was also verified by NMR. Therefore, the transformation process of Cu^{II} to Cu^I was much faster. In contrast, due to the presence of the olefin, we propose Cu^{II}-CF₃ will transfer the -CF₃ to the olefin and regenerate the Cu^I, indicating that it had additional transformation processes of Cu^I to Cu^{II}, which made the Cu^{II} EPR signal dropped slower.

2) In the propose mechanism, Cu(I) is the resting state of Cu, which then is turn into Cu(II)-CF₃, what is the initiation step to convert the CuCl₂ to the active Cu(I). Does the OsII*/I photoredox potential can reduce Cu(II) to Cu(I)? if so, why did the stern-volmer show no quenching in presence of CuCl₂?

Response: We appreciate the comment from the reviewer. In the proposed mechanism, we directly proposed that the Cu(II) was reduced to Cu(I) *via* singlet electron reduction. This electron could be provided by the process of Os(II)* to Os(III) or Os(I) to Os(II). Therefore, we measured the electrochemical redox potential of CuCl₂. For this purpose, 0.1 mmol of CuCl₂ was dissolved in 20 mL 0.1 M tetra-*n*-butylammonium hexafluorophosphate (ⁿBu₄NPF₆) in dry and degassed DCM. Reductions were measured by scanning potentials in the negative direction and oxidations in the positive direction. The obtained value was referenced to Ag/AgCl (3 M KCl).

The reduction potential of CuCl₂ is ca. -0.79 V vs. Ag/AgCl (3 M KCl). Compared to the redox potentials of Os-catalyst in this reaction, Cu(II) cannot be reduced by Os(II)* to obtain the Cu(I) and Os(III) catalyst, but Cu(II) can be reduced by Os(I):

Therefore, Cu(II) cannot quench the excited state of the Os-catalyst. In addition, we found that several publications assumed the Cu(I) was obtained *via* the disproportionation of Cu(II), which could also

explain the formation of Cu(I) not *via* the reduction process from the excited state of the Os-catalyst. We thank the reviewer for the discussion of this part, and we have added the information regarding the formation of Cu(I) in the manuscript and the SI.

Related publications for disproportionation of Cu(II):

Liu, Z., Xiao, H., Zhang, B., Shen, H. *et al. Angew. Chem., Int. Ed.* **58**, 2510–2513 (2019).

Kornfilt, D. J. P., MacMillan, D. W. C. *J. Am. Chem. Soc.* **141**, 6853–6858 (2019).

Sarver, P.J., Bacauanu, V., Schultz, D.M. *et al. Nat. Chem.* **12**, 459–467 (2020).

Ribas, X., Jackson, D. A., Donnadieu, B. *et al. Angew. Chem. Int. Ed.* **41**, 2991–2994 (2002).

3) Also in the mechanistic study, where does the PhCF₃ come from in experiments C and D?

Response: The PhCF₃ was added as an internal standard in the system, since we had to determine quantitatively the amount of potential complexes if they are formed under conditions.

4) The author use [Ru(Bpz)₃][PF₆]₂ as PC for blue light to highlight the advantage of red light. It would be useful to add, in the text or ESI, the redox potential of this PC compared to the reactant (similar to Figure 1C). Is the lack of regioselectivity due to the higher energy blue light or simply because the thermodynamic parameters of the Ru PC don't match as well. Along that line using, another Red-light PC that have different redox potential could be interesting to look at in the future (for example DMQA+ see: *J. Am. Chem. Soc.* 2020, 142, 28, 12056–12061 and *J. Am. Chem. Soc.* 2024, 146, 12, 7922–7930, these articles could be cited in the intro with ref 6-9 for the first one and 12-14 for the second).

Response: We thank and agree with the reviewer regarding the comment about the comparison of the red light and blue light systems. Indeed, since the crucial combination of the photocatalyst, sulfinate salt and -CF₃ reagent has also been determined followed by the thermodynamic parameters, no side product such as undesired β-substituted trifluoromethylated byproduct should be obtained. We agree that the stronger blue light could result in the generation of free -CF₃ radical, causing the direct addition reaction of -CF₃ to the styrene. We have added more explanation to address the difference in regioselectivity in the manuscript.

We also appreciate the suggestion to cite appropriate references. We agree that more potential red-light photocatalysts could also work as long as they are fitting the thermodynamic parameters. We will be glad to observe that more red-light photocatalysts and red light-mediated photocatalytic systems could be explored in the future based on this work.

5) Minor typo:

- In Figure 1, NaSO₂Ph is shown in equiv., while everything else is in mol. Then in Figure 2, NaSO₂Ph is shown in mol%. I would suggest to be consistent and add mol% in Figure 1.

- Text in the mechanistic study need some review, for example:

o "In Figure 5c, it demonstrated..." should be "In Figure 5c, it is demonstrated..."

o "Furthermore, the form of Cu-CF₃ species ..." should be "Furthermore, the formation of Cu-CF₃ species ..."

o "Initially, we attempted to detect the active species in the absence of styrene under model reaction conditions, while no new peak corresponding to CuII-CF₃ was observed in 1 - 4 h, however, we observed the presence of the CuIII(CF₃)₄ anion peak" may read better as "Initially, we attempted to detect the active species in the absence of styrene under model reaction conditions. No new peak corresponding to CuII-CF₃ was observed in 1 - 4 h, however, we observed the presence of the CuIII(CF₃)₄ anion peak"

Response: We thank reviewer's comments. All the typos have been corrected/modified.

Reviewer #2:

The report by Das and coworkers details a practical platform for regioselective sulfonyltrifluoromethylation of alkenes by leveraging red light-mediated photoredox/copper catalysis, enabling the installation of the CF₃ group at internal positions of alkenes. This is an interesting approach using red light to reverse regioselectivity, complementing the well-known regioselectivity of radical-based trifluoromethyl-functionalization of alkenes (Angew. Chem., Int. Ed. 2015, 54, 6999). This chemistry shows a good substrate scope and synthetic applications. Overall, this is a good addition to the red light-mediated photoredox catalysis, although the scope of this chemistry could be further expanded. This reviewer thinks this paper would merit its publication in Nature Communications only if the authors can address the following issues:

Response: We thank the reviewer for their overall positive comments on our work as well as the suggestions that will improve our manuscript. The responses to reviewer's comments are provided below.

1, Unactivated alkenes are amenable substrates in Li's aminotrifluoromethylation reactions (J. Am. Chem. Soc. 2019, 141, 29, 11440). However, this paper only shows the reactivity of styrenes. Have the authors tried unactivated alkenes?

Response: Yes, we have also investigated the unactivated alkenes, however, this strategy is incompatible with unactivated alkenes at this point. The reaction conditions have undergone extensive optimization to achieve regioselective difunctionalization of unactivated alkenes. Unfortunately, we only observed undesired side products rather than the desired product. We assumed that after the addition of -SO₂R radical to the alkenes to form the corresponding carbon-centered radical, the unsuccessful cross-coupling between carbon-centered radical and Cu-CF₃ was the reason of this limitation because we observed hydrosulfonylated product in certain instances. Our group is continuously focusing on the difunctionalization of unactivated alkenes, and we are looking forward to further expanding the substrate scope.

We thank the comment from the reviewer, the unsuccessful substrates we have investigated have been added in **SI 1.4.5**.

2, Although introducing the sulfonyl group is important, have the authors tried other radical precursors? Can we expect similar reactivity if the redox potential of a reagent falls within the window shown in Figure 1? It's important to let the audiences know this approach's synthetic potential.

Response: We haven't tried other radical precursors so far, but we believe that other radical precursors could be used to activate the alkenes and should show similar reactivity as sulfonyl group. To achieve the goal, the radical precursors should be well-selected as we explained in **Figure 1**. In addition, it is imperative to assess the potential of the generated radicals to undergo direct cross-coupling with the Cu-CF₃ complex. For example, several publications have shown the cross-coupling reaction between Cu-CF₃ and alkyl radicals. Therefore, the potential side products could be formed.

In our reaction, the reasons we have chosen sulfinate salts as radical precursors are shown below:

- 1) Sulfonyl groups generated from sulfinate salts are regarded as valuable functional groups in organic and pharmaceutical fields.
- 2) Since we would like to use electrophilic -CF₃ source to form the -CF₃ radical, sulfinate salts as the reductive quenchers are well-fit in the reaction designing.
- 3) As discussed, to avoid undesired side products, radicals employed for alkene activation are anticipated to avoid direct cross-coupling with the Cu-CF₃ complex. Therefore, sulfonyl radicals generated from sulfinate salts are good candidates on this request.

- 4) The process to generate sulfonyl radicals from sulfinate salts is thermodynamically easier because the oxidation potentials of sulfinate salts are not very positive. Therefore, we could use mild photocatalyst, e.g. red-light photocatalyst to realize the reaction.

We thank the comment from reviewer and the utilization of various radical precursors could be advantageous provided their thorough evaluation, thereby broadening the application scope of this approach.

Related publications for the cross-coupling reaction between Cu-CF₃ and alkyl or aryl radicals:

Shen, H., Liu, Z., Zhang, P. *et al. J. Am. Chem. Soc.* **139**, 9843–9846 (2017).

Kornfilt, D. J. P., MacMillan, D. W. C. *J. Am. Chem. Soc.* **141**, 6853–6858 (2019).

Sarver, P.J., Bacauanu, V., Schultz, D.M. *et al. Nat. Chem.* **12**, 459–467 (2020).

3, In Figure 2, it is not meticulous enough to conduct the parallel reactions with only Umemoto's reagent under blue light, as CF₃-source also plays an important role in this reaction (see in Table S2 and part 1.3.2 in SI).

Response: We thank the comment from the reviewer.

To realize the reaction under blue light, 4-vinyl-1,1'-biphenyl, Umemoto's reagent and sodium benzenesulfinate (NaSO₂Ph) were initially chosen as the model substrate, electrophilic -CF₃ source and sulfinate, respectively. Several commercially available Ru- and Ir-based photocatalysts and organic dye, 4-CzIPN, were investigated and all of them offered the desired product in 18–41% yield (**SI 1.3.2**).

Since the [Ru(bpz)₃](PF₆)₂ has provided the best result, we continuously used it as the photocatalyst and further examined the -CF₃ source. Various electrophilic -CF₃ sources such as Mes-Umemoto's reagent, Togni's reagent I/II and Cu(CF₃)₃bpy were utilized under the conditions, however, none of them offered the better yield. The Mes-Umemoto's reagent has more negative reduction potential compared to that of Umemoto's reagent, therefore, the reduction process of -CF₃ source was probably not efficient by [Ru(bpz)₃](PF₆)₂. It is worth noting that both Togni's reagents are able to react with Cu-salts (here was CuCl₂) to form the Cu-CF₃ complexes, which can directly proceed the addition reaction with olefins. Therefore, main side-products were β-CF₃-difunctionalized products. The use of complex bpyCu(CF₃)₃ did not show any conversion of the substrate under this conditions (**SI 1.3.2**).

We chose Umemoto's reagent under blue light in **Figure 2** because it provided the highest yield under blue light after optimizations. And the reason for lower yield compared to “red light” system is the worse regioselectivity, which causing intricate side products as shown in **Figure 2c**. Not only the reactions under blue light with Umemoto's reagent but also with other -CF₃ reagents, all of them showed peaks of side products or even no peak of desired product.

We appreciate the reviewer's comment and have added those explanation in the **SI 1.3.2**.

4, The synthetic utility, which is mainly based on α-trifluoromethyl styrene via elimination of the sulfonyl group, is not very convincing, as α-trifluoromethyl styrene can be easily synthesized via Wittig reaction of trifluoromethylketones. More interesting applications are suggested to be included.

Response: We agree with the reviewer that α-trifluoromethyl styrene derivatives can be synthesized *via* Wittig reaction from trifluoromethylketones or *via* transition metal-catalyzed cross-coupling reactions. Compared to these methods, our approach facilitated the direct synthesis of α-trifluoromethyl styrene derivatives from styrene, circumventing the use of Wittig reagents, as well as -borylated or -halide reagents in the processes, thereby enhancing atom economy. Moreover, generating α-trifluoromethyl styrenes within intricate molecules, which may contain multiple ketone functional groups, through the transformation of ketones to alkenes via the Wittig reaction may lead to the formation of undesired byproducts. Our method offers a supplementary method for synthesizing α-trifluoromethyl styrene derivatives.

We also agree with the reviewer that it will be more interesting that more applications could be provided. As the critical importance of both functional groups, namely -CF₃ and -SO₂R, they will remain intact in our applied application. Therefore, we applied Suzuki-coupling with our product **6** and *p*-tolylboronic acid, and the reaction is shown below:

We have also included the reaction in the manuscript in **Figure 4b**.

5, The Authors proposed an out-of-cage CF₃ radical in Figure 5e. This is consistent with its unique regioselectivity, as CF₃ radicals are prone to adding to styrenes. More experiments and explanations on the selectivity should be provided.

Response: From the control experiments in **Table S3** entry 6, no product could be found in the absence of CuCl₂. However, β-trifluoromethylated products (-CF₃ added at terminal position) could be observed in crude NMR, which indicated the formation or presence of free -CF₃ radicals. As we discussed in the introduction, to circumvent the direct addition of -CF₃ radical to the alkenes, no free -CF₃ radical is the key prerequisite. To achieve this, it is essential to ensure that the generation of sulfonyl radicals precedes the generation of -CF₃ radicals, which can be controlled by thermodynamic parameters. Additionally, the utilization of copper salt could help to capture the unbound -CF₃ radical, leading to the formation of Cu-CF₃ adducts, thereby preventing the presence of free -CF₃ radicals.

6, It would be more appropriate to reflect the involvement of copper catalysis in the title.

Response: We thank the comment from the reviewer. The title of the manuscript has been modified. "Red Light-Mediated Copper-Catalyzed Photoredox Catalysis Promotes Regioselective Switch in the Difunctionalization of Alkenes"

Reviewer #3:

Das et al described an efficient method for the synthesis of sulfonyltrifluoromethylated products through Red Light-Mediated Photoredox Catalysis using Os(btpy)₂(PF₆)₂ as a photoredox catalyst. A series of sulfonyltrifluoromethylated products were synthesized in good yields. The manuscript is well-written and adequate experiments were carried out to support the proposed mechanism. The manuscript can be acceptable for publication after addressing the following queries.

Response: We thank the reviewer for the positive evaluation of our manuscript. The responses to reviewer's comments are provided below.

Is it a radical chain process? Therefore, the quantum efficiency of the reaction must be determined as recently described by Yoon and Stephenson (Science, 2015, 349, 1285).

Response: We appreciate the suggestion from reviewer #3. The quantum yield of the reaction has been investigated and the calculation has also been added in the **SI 1.4.4**.

The light intensity of LED ($\lambda = 650 \text{ nm}$) was 0.02 W/cm^2 . QY for photocatalytic production of **1** was calculated using the following formula¹:

$$\Phi = \frac{N_e}{N_p} \times 100 \% = \frac{2 \times M \times N_a}{\frac{S \times P \times t}{h \times \frac{c}{\lambda}}} \times 100 \% = \frac{2 \times M \times N_a \times h \times c}{S \times P \times t \times \lambda} \times 100 \%$$

Where, M represents the amount of formed **1** (mol), N_a is the Avogadro constant $6.02 \times 10^{23} \text{ mol}^{-1}$, h is the Planck constant $6.63 \times 10^{-34} \text{ J}\cdot\text{s}$, c is the light speed $3 \times 10^8 \text{ m}\cdot\text{s}^{-1}$, S is the irradiation area, which is 0.88 cm^2 , P is the light intensity ($\text{W}\cdot\text{cm}^{-2}$), t is irradiation time (s), λ is the wavelength of light (m).

After calculation, the quantum yield (Φ) is 6.4%, which is much higher than 1. Therefore, this reaction process is supposed to a closed catalytic cycle.²

¹Zhang, T., Schilling, W., Khan, S. U., Ching, H. Y. V., Lu, C., Chen, J., Jaworski, A., Barcaro, G., Monti, S., De Wael, K., Slabon, A., Das, S. *ACS Catal.* **11**, 14087–14101 (2021).

²Kärkäs, M. D., Matsuura, B. S., Stephenson, C. R. J. *Science*, **349**, 1285–1286 (2015).

2. TEMPO was added to intercept the radical reaction. If so, the yield and structure of the adduct should be given.

Response: We agree with the comment from the reviewer. Ideally, we should obtain the yield and structure of the adduct. Unfortunately, the proposed adduct in the manuscript is unstable and the yield cannot be determined. However, the mass of the proposed adduct has been obtained in the HRMS (**Figure 5a**). Therefore, we assumed that the cross-coupling reaction between carbon-centered radical and Cu-CF₃ complexes was impeded by TEMPO, which indicating that the radical process was involved. In addition, the radical probe experiment was also carried out to support our assumption that the ring-opening product was formed under our reaction conditions.

3. The mechanistic proposal provided by the authors, requires better experimental support, such as Stern-Volmer studies.

Response: We agree and appreciate the reviewer that more experimental support for the mechanism is essential. We analyzed the radical processes *via* quenching and radical probe experiments. The fluorescence quenching experiments have also been done and Stern-Volmer plot was prepared (**Figure 5c**), which is corroborated by the electrochemical measurements for redox potentials. The analysis of Cu-CF₃ species is also investigated and we have carried out the EPR analysis, in which we proposed the Cu(II)-CF₃ could be the possible active species.

REVIEWERS' COMMENTS

Reviewer #1 (Remarks to the Author):

The authors have addressed my comments and concerns. I support publication of this manuscript in Nature Communications.

Reviewer #2 (Remarks to the Author):

The authors have effectively addressed all the concerns and comments I raised previously. I believe the paper is now prepared for publication.

Reviewer #3 (Remarks to the Author):

The author has made detailed modifications to the paper based on the reviewer's comments, and I suggest it be accepted for publication